# An eight-year follow-up on auditory outcomes after neonatal hearing screening

Jolien J. G. Kleinhuis[1]*, Karin de Graaff-Korf[2‡], Henrica L. M. van Straaten[2‡], Paula van Dommelen[3‡], Michel R. Benard[1]

1 Pento Speech and Hearing Centers, Leeuwarden, The Netherlands, 2 Dept. Neonatology, Isala Clinics, Zwolle, The Netherlands, 3 Department of Child Health, The Netherlands Organization for Applied Scientific Research TNO, Leiden, The Netherlands

☺ These authors contributed equally to this work.
‡ KGK, HLMS and PD also contributed equally to this work.
* j.kleinhuis@pento.nl

## Abstract

### Objective

The aim of this study is to assess the neonatal click Auditory Brainstem Response (ABR) results in relation to the subsequently determined mean hearing loss (HL) over 1, 2 and 4 kHz, as well as over 2 and 4 kHz.

### Methods

Between 2004–2009, follow-up data were collected from Visual Reinforcement Audiometry (VRA) at 1 and 2 years and playaudiometry at 4 and 8 years of newborns who had failed neonatal hearing screening in the well-baby clinics and who had been referred to a single Speech and Hearing center. Hearing Level data were compared with ABR threshold-levels established during the first months of life. The Two One-Sided Tests equivalence procedure for paired means was applied, using a region of similarity equal to 10 dB.

### Results

Initially, in 135 out of 172 children referred for diagnostic procedures hearing loss was confirmed in the neonatal period. In 106/135 of the HL children the eight-year follow-up was completed. Permanent conductive HL was established in 5/106 cases; the hearing thresholds were predominantly stable over time. Temporary conductive HL was found in 48/106 cases and the loss disappeared by 4 years of age at the latest. Sensorineural hearing loss (SNHL) was found in 53/106 cases, of which 13 were unilateral and 40 bilateral. ABR levels were equivalent (within a 10 dB range) to VRA levels at age 1 and 2 and play audiometry levels at age 4 and 8, both when VRA and play audiometry were averaged over both frequency ranges.

### Conclusion

Long term follow-up data of children with SNHL suggest that the initial click ABR level established in the first months of life, are equivalent to the hearing threshold measured at the age

**Data Availability Statement:** All relevant data are within the paper.

**Funding:** The authors received no specific funding for this work.

**Competing interests:** The authors have declared that no competing interests exist.

of 1, 2, 4 and 8 years for both mean frequency ranges. Click ABR can reliably be used as starting point for long-term hearing rehabilitation.

## Introduction

Hearing loss is found in approximately one out of thousand newborns [1–3]. Early intervention in neonatal established hearing loss (HL) promotes the language and social-emotional development of children with HL [4] and is shown to be most effective when initiated as early as possible [5]. Rehabilitation at an early age with hearing aids, cochlear implants and/or bone-conduction devices can result in near-normal speech and language development, as well as improving school performance, self-esteem and psychosocial adaption. This is in contrast with results on these markers if intervention takes place later in life, demonstrating the importance of early detection of HL [6–8].

To improve the development of children with HL at an early age, a two-step Automated Auditory Brainstem Response (AABR) neonatal hearing screening program was introduced in 1998 for NICU newborns in the Netherlands [9]. This was followed by a nationwide universal hearing screening for (near) term well-babies in 2004 [10, 11]. In this Dutch well-baby program, hearing screening is offered during home visits between the 4th and 7th day after birth. It consists of an initial oto-acoustic emission (OAE) measurement. At a refer on this test, a second measurement is performed a few days later. If the second measurement does not provide satisfactory responses, a third attempt is made through AABR [12]. In the case of an unsatisfactory response, hearing diagnostics in a regional speech and hearing center is performed through Auditory Brainstem Response (ABR), OAE and impedance audiometry to establish the severity and type of HL. If HL is established, hearing rehabilitation is provided at this center. Data from the initial ABR measurements form the basis for hearing rehabilitation by translating the measurements into estimated hearing thresholds using international standards, to be used, for example, for hearing aid fitting [13].

Acquiring ABR data can be performed with acoustic clicks, tone pip, tone burst, or noise burst as stimuli [14–17]. At the time the ABR-measurements of this analysis were performed (between 2004 and 2009), the air-conduction click-stimulus was still the gold standard for ABR in the Netherlands [18]. To date, the click-stimulus is still used in some clinics for practical reasons, e.g., to reduce testing time or to diagnose auditory neuropathy [19, 20]. Several studies showed the correlation of the predictive value of early established ABR findings with Pure Tone Average (PTA) results at an older age [15, 16, 19, 21–24]. Besides good correlations between click ABR and PTA, click stimuli also caused controversy. For example, Gorga (2006) [19] found that, depending on the degree of hearing loss, the click-ABR underestimated the eventual hearing threshold. The click-ABR is shown to be an adequate indicator of the type of hearing loss in babies referred from newborn hearing screening, however, the predicted value is reduced in some neonatal groups [21]. Cheng (2021) [25] supports this observation and suggests caution in using click-ABR to estimate PTA in children. Frequency-specific stimuli is shown to be more adequate for the fitting of hearing aids and is considered as the gold standard nowadays [13, 20].

In this study we evaluated the early established ABR findings over time. Click-ABR appeared to be related most closely to the audiometric thresholds at 2 and 4 kHz, with relatively poor agreement at 1 kHz [26]. However, Lu (2017) [20] found similar correlations between click-ABR and PTA thresholds at 1–4 and 2–4 kHz. Therefore, we analyzed the

thresholds predicted by the click-ABR to examine whether the final mean hearing threshold can be best predicted by the wide frequency range 1–4 kHz or by the narrower 2–4 kHz range at 8 years of age.

Furthermore, this study assesses the initial ABR results obtained in the first months of life In comparison to the outcomes of Visual Reinforcement Audiometry (VRA) at the age of 1 and 2 years and play audiometry at the age of 4 and 8 years. It is scientifically interesting to evaluate to what extent the established neonatal hearing thresholds are predictive in the long term, moreover since the click-ABR is currently still used in some clinics [20, 25]. Even to date, this analysis is clinically relevant for audiologists to inform caregivers of babies with HL about the impact of this hearing loss in the immediate future as well as later on.

## Methods

This single-center retrospective study was performed on data collected from (near) term children born in the period 2004–2009 who were referred to our speech and hearing center in the Netherlands. For children with HL, the type and severity of their HL was verified through age-appropriate tests throughout the child's life. Follow-up audiologic data were collected and analyzed from first referral ($<$ 3 months of age) until 8 years of age. Data were anonymized upon collection, so that the authors did not have access to information that could identify individual participants during or after analysis. For this study, a distinction was made between temporary conductive HL, permanent conductive HL, Auditory Neuropathy Spectrum Disorder (ANSD) and sensorineural hearing loss (SNHL). The initial ABR measurements of children with SNHL were utilized to assess the ABR data in relation to hearing thresholds measured at ages 1, 2, 4 and 8 years. The ABR data are compared to the hearing thresholds averaged over 1, 2 and 4 kHz as well as averaged over 2 and 4 kHz.

### Auditory diagnostic tests

Several auditory hearing tests were used, depending on the age and development of the children with HL. The outcomes of these assessments were used for the long-term follow-up.

**Auditory Brainstem Response measurements.** With ABR, the neural activity of the cochlear nerve as a response to acoustic stimuli is recorded [27]. Acoustic stimuli are provided through insert-earphones. The acoustic stimuli consist of clicks. The neural activity is detected by electrodes placed on the forehead, as close to the vertex as possible and behind both ears of the infant, with reference to a ground electrode on the cheek or temple. The electrode impedances were pursued to be roughly equivalent and $<$ 5 kOhms throughout the test. The intensity of the stimulus is limited to 100 dB nHL. In most cases, a stimulus up to 80 dB nHL is used. The responses were filtered between 100 and 3000 Hz, and the stimulation rate was set at 31.1 clicks/sec. The ABR thresholds are defined as the lowest level at which a clear response is present, with a response absent at a level 5 or 10 dB below the threshold, obtained under good recording conditions, as described in the NHSP guidance for ABR-testing in babies [13]. When a hearing loss is found, a second ABR measurement is offered as a control and as a check for (no) fast progression in the hearing loss. All but five children with SNHL had at least 2 ABR measurements, 10 children had 3 ABR measurements. The Interacoustics' Eclipse EP15 (Interacoustics A/S, Middel-fart, Denmark) is used to measure the air-conduction ABR responses. No bone-conduction ABR measurements are performed. The equipment is calibrated according to the advised calibration standards, as is specified in its manual.

**Visual Reinforcement Audiometry.** When infants can turn their head, they can be tested with Visual Reinforcement Audiometry [28]. VRA is based on the orientation reflex towards a sound source, requires cooperation of the child and aims to achieve a conditioned reflex from

the child. VRA can best be performed on children between 6 and 24 months of developmental age [29]. The child is conditioned to respond to an acoustic stimulus. If the child turns its head towards the sound, its attention will be visually reinforced by colored lights, rotating 3D toys or a dancing teddy bear behind tinted acrylic glass. VRA determines the minimal loudness of sounds in the frequency range of at least 0.5–4 kHz to which the child reacts repeatedly. The stimuli can be provided in free field, or by headphones/insert earphones, to test each ear individually. The children included in our study were all tested through free field stimulation, because it was the standard at this speech and hearing center at that time. The test protocol used is based on the recommended procedure for VRA by the British Society of Audiology [30]. The starting point is a stimulus of 1 kHz at 70 dB HL. Then the frequency is varied, keeping the loudness at 70 dB HL until the child is conditioned. Next, the stimulus is decreased by 10 dB, still switching between frequencies, each time a clear reaction is found. When there is no clear reaction, the stimulus is increased with 10 dB on this frequency. The threshold is set at the lowest level at which a clear reaction is found, preferably with reproducibility. Interacoustics' Affinity / AC440 is used, calibrated to ANSI S3.22–1996 Standard, to present the acoustic stimuli.

**Play audiometry.** During play audiometry, the examined child is taught to perform a small task when it hears the presented acoustic stimulus. This task is best suited for children between 2 and 6 years of age. If the child is older, he/she may prefer to press a button on hearing the acoustic stimuli. In that case we speak of "pure-tone audiometry". Both play and pure tone audiometry determine the minimal loudness of sounds in the frequency range of at least 0.5–4 kHz at which the child perceives the stimulus. The acoustic stimuli are provided in the free field, via headphones or insert-earphones, or by bone-conductor. In our study, most children were tested via headphones (TDH39), because this was standard practice at our center at the time. A few children were tested through free field stimulation as headphones were not accepted (n = 1), or because sufficient hearing (thresholds below 25 dBnHL) was already confirmed through free field play/pure tone audiometry plus OAE's for both ears (n = 7). Interacoustics' Affinity / AC440, calibrated to ANSI S3.22–1996 Standard, is used to present the acoustic stimuli.

## Auditory diagnoses

In our study population, we define the different types of hearing loss as follows:

- Permanent conductive hearing loss is diagnosed when ABR threshold are ≥ 25 dB, middle ear measurements are abnormal, not caused by middle ear fluid, or when middle ear measurements are medically not possible, for example in cases of microtia.

- Temporary conductive hearing loss is diagnosed when initial ABR thresholds are ≥ 25 dB with abnormal middle ear measurements (flat line in tympanometry), and follow-up measurement show normal hearing.

- Sensorineural hearing loss is diagnosed when ABR thresholds are ≥ 25 dB, with normal middle ear measurements and absence of OAE's.

- Auditory Neuropathy Spectrum Disorder (ANSD) is diagnosed when there is an absence of normal ABR patterns, cochlear microphonics are observed in the ABR signal and presence of OAE's.

- Hearing loss is found progressive when, according to Dutch field standards ("Veldnorm hoortoestelverstrekking", 2013), the hearing loss has increased at least 5 dB for 4 frequencies,

or 10 dB for 3 frequencies, or 15 dB for 2 frequencies, or 20 dB for 1 frequency, compared to the previous measurement.

## Statistical analysis

For this study, the mean difference between the ABR levels and VRA or play audiometry was investigated to determine whether these results were equivalent (differ at most by a small pre-defined amount). The paired TOST (two one-sided tests) is a test of equivalence based on the classical t-test used to test the hypothesis of equality between two paired means. It was applied with the magnitude of the region of similarity equal to 10 dB [31]. The null hypothesis was that the differences in mean dB levels for the different tests were not equal. The overall P-value of the equivalence test is taken as the larger of the two P-values. P-values <0.05 were considered significant. Note that TOST reverses the roles of the null and alternative hypotheses in a two-sided t-test. Furthermore, Bland-Altman plots were calculated to show the agreement between ABR and the follow-up tests. For each follow-up test, the Bland-Altman plot shows the mean differences of VRA1, VRA2, PTA4, PTA8 with ABR (the 'bias'), and 95% limits of agreement as the mean difference. For each individual, the difference of ABR and the follow-up test are plotted against the average of ABR and the follow-up test. R Version 3.6.2 with library equivalence was used for statistical analysis.

The statistical analysis showed that the intraclass correlation coefficient of both ears in bilateral HL was 0.6. This means that the degree of HL in one ear is too strongly correlated with the degree of HL in the other ear per child with HL. Therefore, for all children, only one of the two ears was selected and further analyzed, namely, the ear with the greatest HL. For unilateral hearing loss, this means the only ear with hearing loss is selected and for bilateral hearing loss the ear with the greatest hearing loss is selected. When more than one ABR measurement was available we used the most representative ABR measurement (with the least hearing loss) for the analysis. This is because during some ABR test moments, there was an (additional) hearing loss because of middle ear fluid. If we take the most favorable ABR measurement, we look at the purest sensorineural hearing loss of the child.

## Ethics

This study was approved by a Medical Ethics Committee of the Isala Hospital, Zwolle, The Netherlands, reference number 220101.

## Results

In the study period, 172 children were referred to our speech and hearing center (Fig 1). At the first visit, within the first 3 months of life, HL was confirmed in 135 (N = 135/172) children, 35 (N = 35/172) had normal hearing and 2 (N = 2/172) did not accept the invitations for the auditory assessments. Of the 135 children with HL, 29 (N = 29/135) had incomplete follow-up data: they had moved to another city, were referred for cochlear implants, they did not want a follow-up or they died, leaving 106 children with eight years of auditory follow-up.

Our threshold data is succinctly presented in Table 1. While this table provides a comprehensive overview of our data, we acknowledge its limited representational value due to the fact that not all children who underwent ABR received audiometric follow-up at each test interval. For instance, in the context of play audiometry at age 8, the data is drawn from a subset of the population covered by the ABR data. As a result, we have displayed the data in a different format in Figs 1–4.

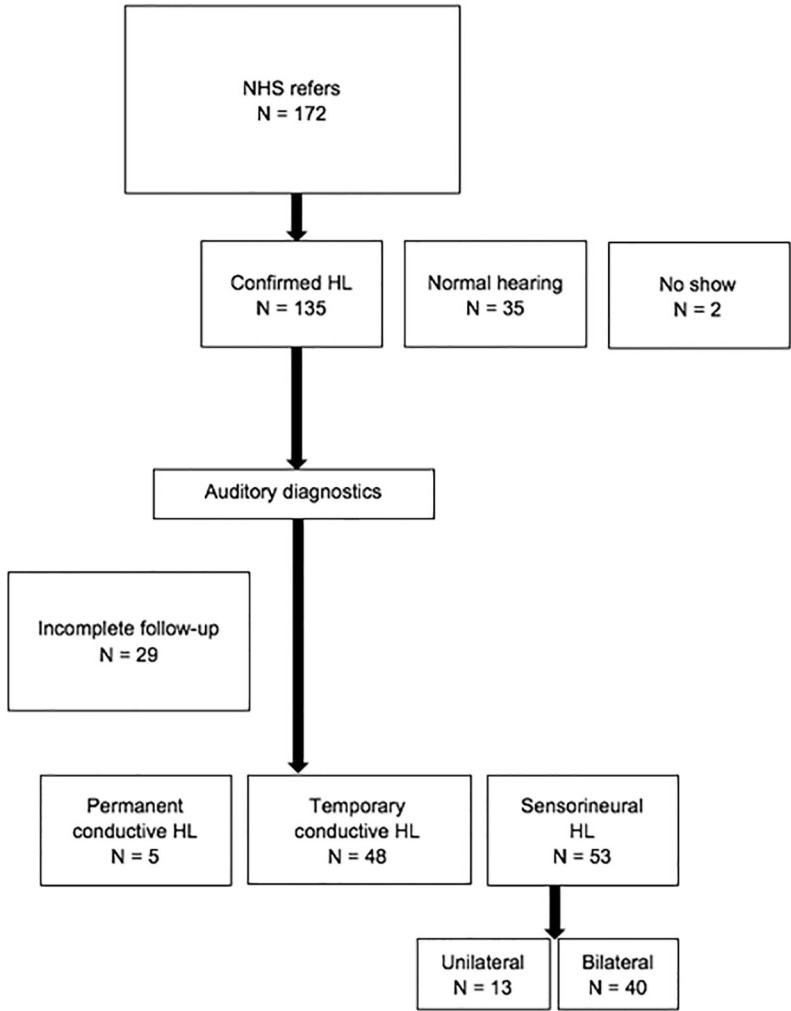

**Fig 1. Flow-chart of children referred to our speech and hearing center with their follow-up route as regards to type of HL.** PCHL = permanent conductive hearing loss, TCHL = temporary conductive hearing loss.

**Table 1. Mean threshold values for all follow-up measurements and for the different types of hearing loss.**

| Audiometric tests → Diagnosis ↓ | ABR | VRA 1 (1–4 kHz) | VRA 1 (2–4 kHz) | VRA 2 (1–4 kHz) | VRA 2 (2–4 kHz) | Play 4 (1–4 kHz) | Play 4 (2–4 kHz) | Play 8 (1–4 kHz) | Play 8 (2–4 kHz) |
|---|---|---|---|---|---|---|---|---|---|
| Temporary CHL | 44 (12) | 32 (12) | 31 (12) | 25 (9) | 24 (8) | 17 (9) | 16 (7) | 15 (6) | 15 (7) |
| Permanent CHL | 65 (12) | 51 (22) | 47 (25) | 31 (15) | 30 (13) | 55 (24) | 52 (23) | 52 (16) | 50 (14) |
| Unilateral SNHL | 77 (16) | 31 (12) | 31 (12) | 23 (15) | 24 (15) | 75 (14) | 74 (15) | 85 (20) | 85 (22) |
| Bilateral SNHL | 69 (21) | 63 (17) | 67 (18) | 63 (17) | 65 (18) | 61 (15) | 61 (15) | 64 (18) | 63 (19) |

This table presents the average threshold values in decibels (dB) for various follow-up tests across different types of hearing loss, along with their corresponding standard deviations in parentheses. The mean thresholds are provided for ABR, VRA at ages 1 and 2 (with perceptual tonal averages at 1–4 kHz and 2–4 kHz), and play audiometry at ages 4 and 8, also including perceptual tonal averages at 1–4 kHz and 2–4 kHz.

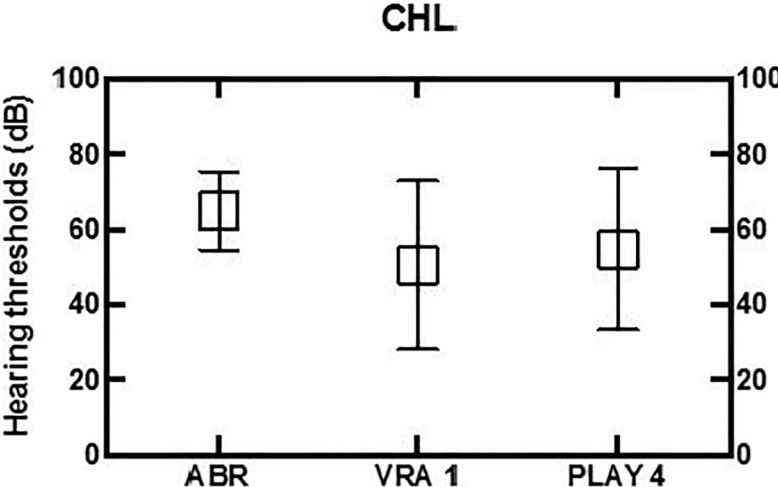

**Fig 2. Follow-up data on permanent conductive HL.** ABR results and play audiometry of the ear with the most HL. VRA is performed in free-field, showing the best ear thresholds.

Of the 106 children (N = 106/135), hearing in five children was assessed as permanent conductive HL (N = 5/106, 2 unilateral and 3 bilateral). Fig 2 shows the permanent conductive HL with a mean ABR of 65 dB nHL, VRA at age 1 of 48 dB and 57 dB at age 4. Hearing thresholds, while persistent, appeared to be unstable, probably due to the varying severity of the middle ear problems for these children (e.g. aural atresia or perforated eardrum).

Fig 3 shows the temporary conductive HL, as it was found in 48 children (N = 48/106) with a mean ABR of 44 dB nHL, decreasing to normal hearing thresholds at 4 years of age at the latest, their middle-ear problems having cleared up by then.

In the 53 cases with SNHL, 40 (N = 40/53) children were diagnosed with bilateral SNHL, and 13 (N = 13/53) children with unilateral HL, with mean 71 dB nHL (*SD* = 20 dB) thresholds on the click ABR for the ear with the greatest HL. For children with established bilateral SNHL, ABR levels were compared and analyzed with the follow-up tests (N = 32 for VRA at

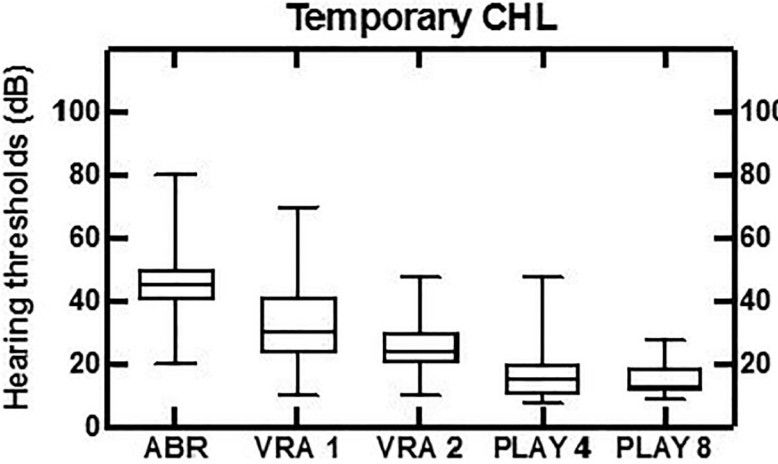

**Fig 3. Data from temporary conductive HL children.**

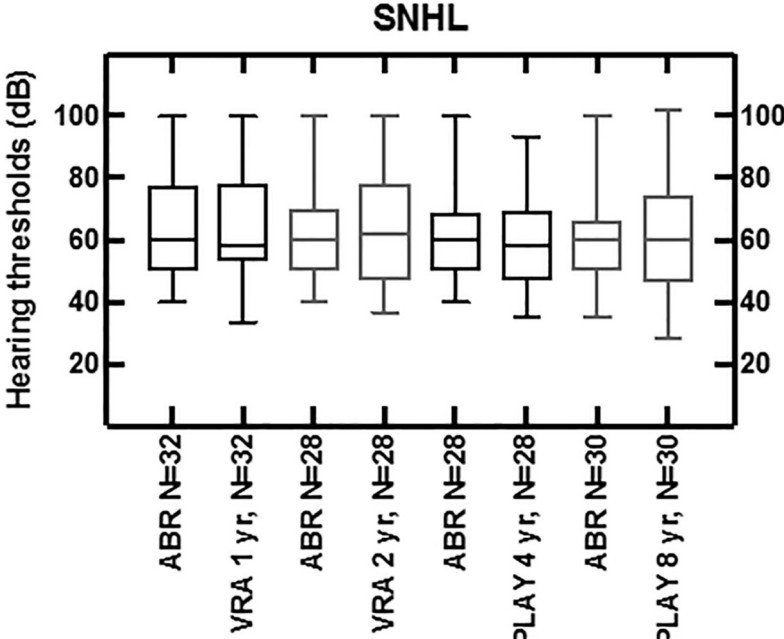

**Fig 4. The initial ABR levels compared to the thresholds in dB of VRA at ages 1 and 2, and play audiometry at ages 4 and 8, averaged over 1, 2 and 4 kHz from the same group of children with HL, missing data excluded.**

age 1; N = 28 for VRA at age 2; N = 28 for play audiometry at age 4; N = 30 for play audiometry at age 8). The reason for having 30 children with follow-up data at the age of 8 instead of 40 is because certain children underwent follow-up at a different institute (N = 2) or were referred for cochlear implants (N = 8). The visualization of Fig 4 leaves out missing data: the initial ABR was compared to the thresholds in dB nHL of VRA at ages 1 and 2, and play audiometry at ages 4 and 8, averaged over 1, 2 and 4 kHz from the same group of children with HL. The difference of the means when compared to the ABR data, averaging with absolute values, are 8 dB for VRA at age 1, and 9 dB for tests at ages 2, 4 and 8.

In the case of children with unilateral SNHL, similar comparisons were conducted, revealing significantly better hearing thresholds at VRA at ages 1 and 2. Hearing thresholds aligned closely with ABR data at ages 4 and 8. This can be reasonably explained by the fact that VRA is conducted in free field conditions, allowing the unaffected ear to participate.

Figs 5–8 show Bland-Altman plots for ABR compared to the different follow-up tests for bilateral HL. These figures show that the mean differences (black line) were close to 0, which implies that the bias is small. The 95% limits of agreement (dotted line) exceeded the magnitude of the region of similarity, which implies that for more than 5% of individuals the difference of ABR and the follow-up test was greater than ± 10 dB nHL.

The statistical analysis showed that the hearing thresholds were equivalent within a 10 dB range for all audiometric test comparisons to the click-ABR levels. The hearing thresholds showed equivalence when data were averaged over 1–4 kHz as well as when averaged over only 2–4 kHz. P-values for all comparisons were ≤ 0.005.

## Discussion

For the counseling of caregivers of children with HL, it is clinically relevant to provide an evidence-based prediction of expected HL in the future directly after neonatal ABR. Therefore,

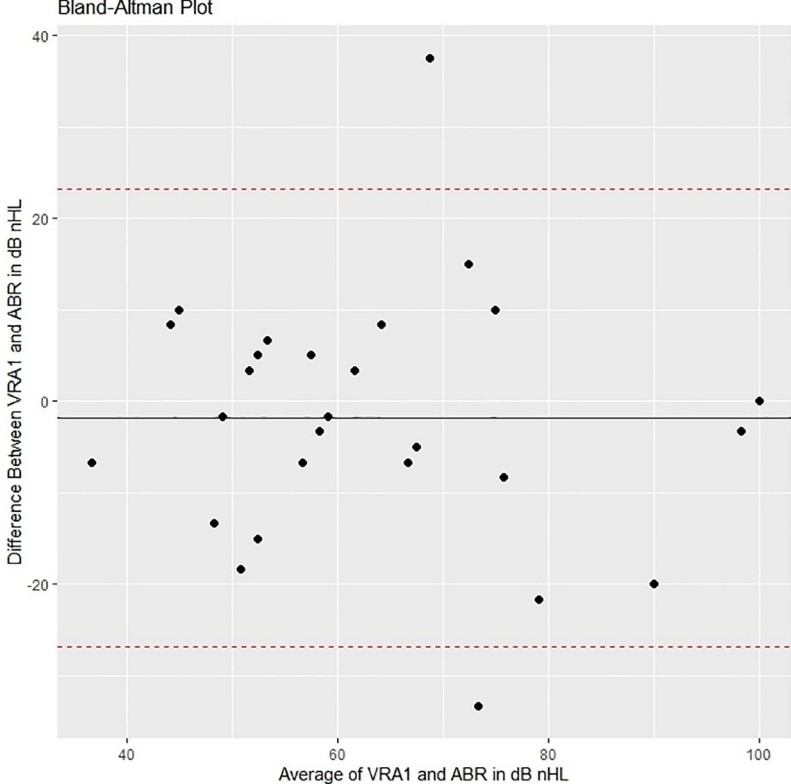

**Fig 5. Bland-Altman plot comparing ABR with VRA at age 1.**

we analyzed our historical clinical data from the last decade. At that time, i.e. between 2004 and 2009, click ABR was still the gold standard in the Netherlands, as it is still in other countries [20, 25]. So to date, click ABR data still have relevance. Our study comprises a single center study, which we see as a benefit when considering equality of calibrations of the used devices, judgement of ABR thresholds and VRA responses. There is already evidence that the click-ABR predicts hearing thresholds in children accurately [15, 16, 19, 21–24]. This study focuses on the value of the initial click ABR in relation to the hearing thresholds measured throughout the child's life until age 8. We also aimed to compare hearing threshold averaged over 1–4 kHz with thresholds averaged over 2–4 kHz, analyzing which of the two is better related to the initial ABR data.

In our study, we found five cases with permanent conductive HL in which the hearing threshold remained stable over time. In 48 cases temporary conductive HL was found but disappeared by age 4 at the latest. In 53 cases SNHL was found: 13 unilateral and 40 bilateral. The ABR levels were equivalent (within a range of 10 dB) with VRA levels at age 1 and 2 and with PTA levels at age 4 and 8.

Standard practice in neonatal hearing assessment is to obtain estimates of hearing thresholds for both low and high frequencies. Nowadays, this is preferably measured using frequency specific ABR measurements and ASSR [13–17]. Once children are old enough to obtain reliable behavioral data (VRA), this data will be used for hearing aid fitting in addition to the ABR results. For the initial fitting of hearing-aids, it is important to understand how the measured ABR thresholds relate to the estimated hearing levels of this very young population. Correction

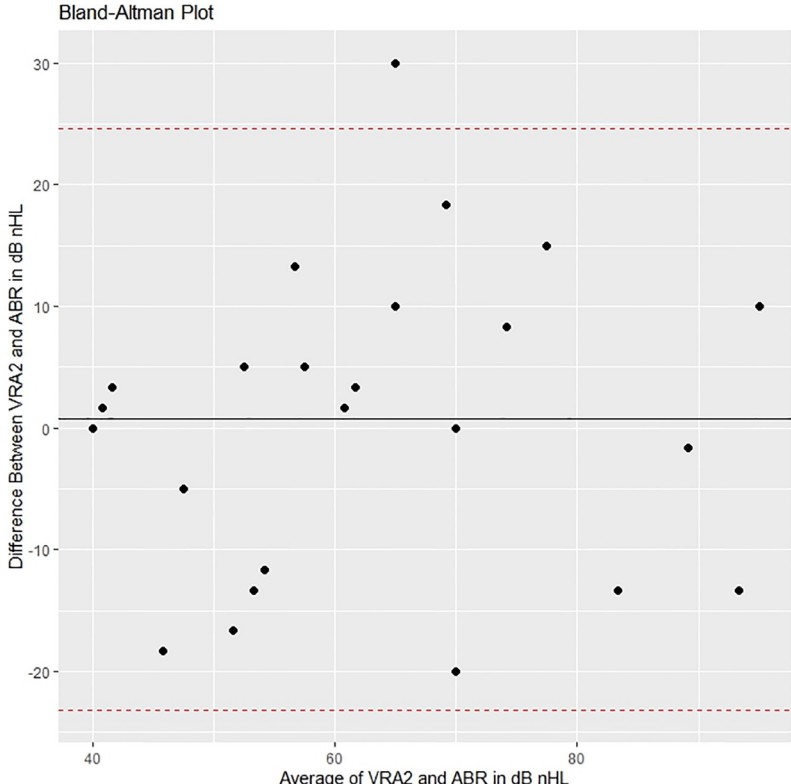

**Fig 6. Bland-Altman plot comparing ABR with VRA at age 2.**

factors that can be used to convert frequency specific ABR thresholds into estimated thresholds for hearing aid fitting, have been published [13].

The results of this study are in line with the auditory eight-year follow-up findings of babies from the neonatal intensive care units [31]. Only two out of 172 children did not accept the diagnostic auditory assessment after neonatal hearing screening. This is a low 'no show' number at the start of a study in comparison to international references [32]. In 135 out of the 170 children HL was found of which 106 completed the full eight-year follow-up. As expected, a large proportion of children with temporary conductive HL recovered from middle ear problems and had normal hearing at follow-up. This study shows that for children with temporary conductive HL, VRA often does not show normal hearing at 1 year of age, with mean hearing thresholds of 30 dB. Normal hearing is established only later with play audiometry at age 4. As the first 2 years are a critical period for speech and language development, these findings emphasize the need of early revalidation of hearing loss, including in children with temporary conductive hearing loss [8]. In children with temporary HL, a longer follow-up period may be required. For the children with permanent conductive HL (although a small number of N = 5), the hearing thresholds showed a slight improvement over time. The VRA is conducted in the free field, and it therefore tests the best ear. Two out of five children with permanent conductive HL had unilateral conductive HL, which explains the larger variation (Fig 2) for the VRA compared to the ABR results.

The present study shows that for children with SNHL the mean initial ABR findings were highly predictive for the HL found at ages 1 and 2 (VRA), and at ages 4 and 8 (play

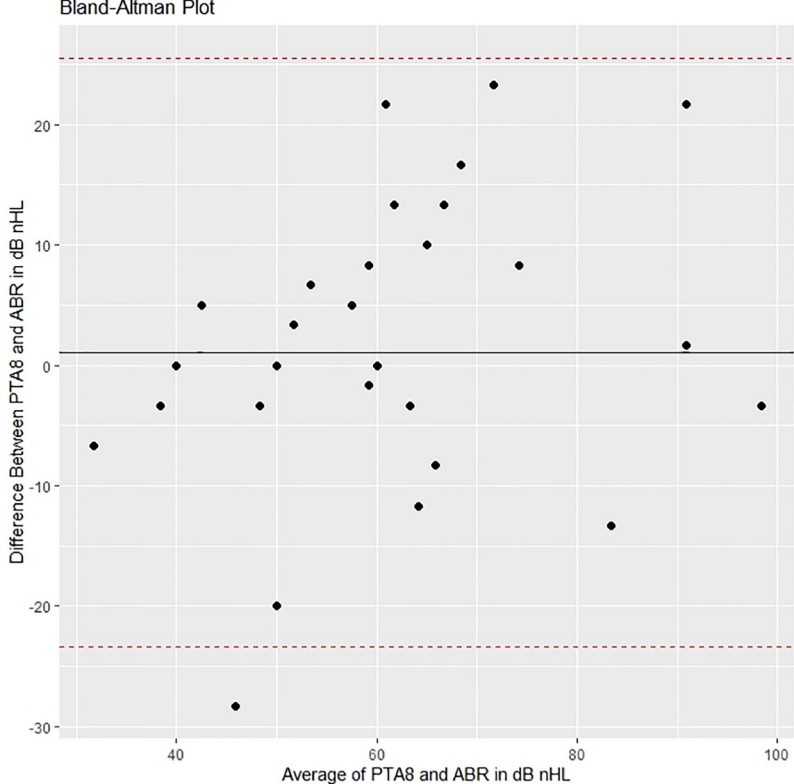

**Fig 7. Bland-Altman plot comparing ABR with PTA at age 4.**

audiometry). Our statistical analysis show that this applies to both the 1–4 kHz and 2–4 kHz range. Earlier studies have mainly shown the predictive value of the click-ABR for the 2–4 kHz range [16, 22, 26]. We found that click-ABR effectively determines both averaging ranges. This highlights the fact that click-ABR can be reliably used as the high Fletcher average for initial hearing aid fitting, as it also accurately predicts the hearing aid threshold averaged over 1–4 kHz.

In this well-baby population, we did not find indications of maturation of the acoustic nerve [33]. Whether or not the stability of HL persists later in life, i.e., after the age of 8 years, is obviously largely dependent on the etiology of the HL, particularly if it is a genetic form of progressive HL. This increases the need to search for the underlying etiology case of SNHL, as it can affect the management and counselling of HL [34, 35].

In this retrospective analysis, no children with auditory neuropathy were identified, although OAE and ABR were recorded in the neonatal hearing assessment. The prevalence of ANSD in well-babies with confirmed HL is reported to be 6.5% [36]. This implies that three to four children (6.5% of 53 hearing impaired children) in our population would be expected to have auditory neuropathy. A reason for this may be that ANSD mainly occurs in premature children [37]. The main reason is probably the fact that hearing screening in the Netherlands is performed with OAE measurements. Children with ANSD do have oto-acoustic emissions and will therefore (falsely) pass the newborn hearing screening. With the Dutch hearing screening performed with OAEs, a number of children with ANSD may therefore be missed.

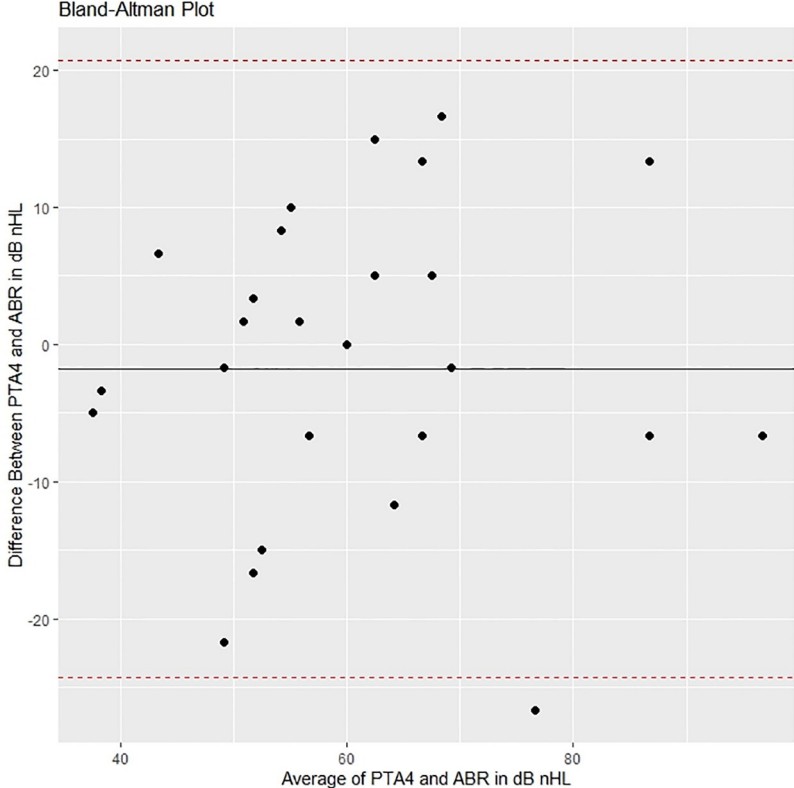

**Fig 8. Bland-Altman plot comparing ABR with PTA at age 8.**

While our investigation explored the predictive significance of initial click ABR on hearing thresholds throughout a child's life until age 8, we recognize that the sample size (N = 30 at age 8) imposes constraints on the generalizability of our findings, and we emphasize the need for caution when extrapolating our findings to broader populations.

A second limitation of our study was the intensity limit of the used stimulus. A number of children with profound HL, who had no identifiable peaks in their ABR traces, were registered with an ABR hearing threshold set as >80 dB. Thresholds up to 120 dB were established in subsequent audiometry tests. For these cases, ABR levels appeared to be non-equivalent with audiometry, but ABR levels could simply not fully establish the degree of HL. However, this limitation did not influence the technical intervention with hearing-aids.

Despite these limitations, long term follow-up data of children with SNHL suggest that the initial click ABR level established in the first months of life are equivalent to the hearing threshold measured at the age of 1, 2, 4 and 8 years for both mean frequency ranges. Click ABR can reliably be used as starting point for long-term hearing rehabilitation.

## Author Contributions

**Conceptualization:** Jolien J. G. Kleinhuis, Karin de Graaff-Korf, Henrica L. M. van Straaten, Michel R. Benard.

**Data curation:** Jolien J. G. Kleinhuis, Paula van Dommelen, Michel R. Benard.

**Formal analysis:** Jolien J. G. Kleinhuis, Paula van Dommelen, Michel R. Benard.

**Investigation:** Jolien J. G. Kleinhuis, Michel R. Benard.

**Methodology:** Jolien J. G. Kleinhuis, Karin de Graaff-Korf, Henrica L. M. van Straaten, Michel R. Benard.

**Project administration:** Jolien J. G. Kleinhuis, Michel R. Benard.

**Resources:** Jolien J. G. Kleinhuis, Paula van Dommelen.

**Software:** Jolien J. G. Kleinhuis, Paula van Dommelen, Michel R. Benard.

**Supervision:** Karin de Graaff-Korf, Henrica L. M. van Straaten, Paula van Dommelen, Michel R. Benard.

**Validation:** Jolien J. G. Kleinhuis, Paula van Dommelen, Michel R. Benard.

**Visualization:** Jolien J. G. Kleinhuis, Paula van Dommelen, Michel R. Benard.

**Writing – original draft:** Jolien J. G. Kleinhuis, Karin de Graaff-Korf, Michel R. Benard.

**Writing – review & editing:** Jolien J. G. Kleinhuis, Karin de Graaff-Korf, Henrica L. M. van Straaten, Paula van Dommelen, Michel R. Benard.

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
