## [Decision Letter · Decision Letter 0]

22 Aug 2023

PONE-D-23-17786The click-ABR as predictor: an eight-year auditory follow-up after neonatal hearing screeningPLOS ONE

Dear Dr. Kleinhuis,

Thank you for submitting your manuscript to PLOS ONE. After careful consideration, we feel that it has merit but does not fully meet PLOS ONE’s publication criteria as it currently stands. Therefore, we invite you to submit a revised version of the manuscript that addresses the points raised during the review process.

We look forward to receiving your revised manuscript.

Kind regards,

Paul Hinckley Delano, Ph.D.

Academic Editor

PLOS ONE

2. Please upload a copy of Supporting Information Figure (S1-S8) which you refer to in your text on page 19.

3. Please keep your tables as part of your main manuscript and remove the individual files. Please note that supplementary tables (should remain/ be uploaded) as separate "supporting information" files

Reviewers' comments:

Reviewer's Responses to Questions

**Comments to the Author**

1. Is the manuscript technically sound, and do the data support the conclusions?

Reviewer #1: Partly

Reviewer #2: Yes

2. Has the statistical analysis been performed appropriately and rigorously? 

Reviewer #1: Yes

Reviewer #2: Yes

3. Have the authors made all data underlying the findings in their manuscript fully available?

Reviewer #1: Yes

Reviewer #2: Yes

4. Is the manuscript presented in an intelligible fashion and written in standard English?

Reviewer #1: Yes

Reviewer #2: Yes

5. Review Comments to the Author

Reviewer #1: Dear authors,

After carefully reviewing your manuscript, I would like to extend my congratulations on your work. Overall, the presented data seems interesting and appropriately collected and analyzed. However, I believe that the main objectives described in the paper are not fully addressed. For instance, the study aims to evaluate the predictive value of the initial click ABR in healthy children regarding their perceptual auditory thresholds throughout their life up to the age of 8 years. Unfortunately, the amount of missing data is significant, and the sample size at 8 years is relatively low. Additionally, it is widely known that there is a relationship between electrophysiological and perceptual thresholds in newborns. If a child's hearing remains unchanged over time, these thresholds will naturally remain equivalent. In such cases, it would be very interesting to have a report on the observed rate of progressive sensorineural hearing loss in your cohort.

The second objective was to determine which perceptual tonal average (1,2, 4 kHz, or 2.4 kHz) best matches the electrophysiological threshold. In this regard, I believe it is essential to contrast the results of the assessments considering the profile of the auditory curve (ascending, descending, flat, among others).

Therefore, to be considered for publication, I recommend reevaluating and refocusing the scope of your work. It would be beneficial to concentrate on describing the results in detail, including the protocols used, and providing a more comprehensive description of the observed epidemiological findings.

A detailed list of the suggestions by topic is given below:

Introduction:

L65: Describe the AABR acronym.

Methods:

It is very important to detail each of the protocols used, referencing them where appropriate.

ABR: please indicate equipment calibration, protocol for obtaining thresholds, derivation of electrodes (positive vertex?).

Visual reinforcement audiometry: Includes theoretical information on what the technique consists of and how it is recorded, but does not detail the protocol used.

Ethics: I think it is important to mention the name of the ethics committee that approved the study protocol.

Results:

I recommend a summary table with the most important data.

L245-246: It would be interesting to know the evolution of unilateral sensorineural hearing loss.

L 250-252: This analysis can confuse the reader because if negative values are averaged with positive values, the result will tend to be 0. I recommend averaging with absolute values. This will better reflect what was observed in the Bland-Altman plots.

Discussion:

I recommend strengthening the limitations and scope section of the study.

L290-293: As noted above, I feel that this statement does not reflect the actual scope of the work. I recommend it be reconsidered.

L256-357: Similar to the previous paragraph.

Reviewer #2: The article provides useful information for physicians treating children with hearing loss. The authors establish the relationships between ABR and audiometry in the first years of life, demonstrating the usefulness of click ABRs in predicting hearing loss PTAs. Only minor changes need be made.

line 65 and 72: put abbreviation for AABR in line 65, not 72.

line 96: either 1 khz? fix this sentence

line 100: ends with ?

line 104: you use PTA as the abbreviation for pure tone audiometry but also for pure tone average, please eliminate this abbreviation for pure tone audiometry, either change it or eliminate the need for abbreviation.

line 127 and 128: eliminate first sentence, not needed

line 129: doble .., eliminate

line 134: 31,1 should be 31.1

line 139: you already established abbreviation for SNHL, add it here

line 158: eliminate parenthesis for (play-)audiometry, not needed

line 174, 177: change aberrant

line 197: eliminate sentence about two sided test, not needed

line 295, change 5 to five

6. PLOS authors have the option to publish the peer review history of their article (what does this mean?). If published, this will include your full peer review and any attached files.

Reviewer #1: No

Reviewer #2: No

---

## [Author Response · Author response to Decision Letter 0]

5 Oct 2023

See the document "Response to Reviewers" for a point by point response to the comments of the editor and reviewers.

---

## [Decision Letter · Decision Letter 1]

29 Nov 2023

PONE-D-23-17786R1The click-ABR as predictor: an eight-year auditory follow-up after neonatal hearing screeningPLOS ONE

Dear Dr. Kleinhuis,

Thank you for submitting your manuscript to PLOS ONE. After careful consideration, we feel that it has merit but does not fully meet PLOS ONE’s publication criteria as it currently stands. Therefore, we invite you to submit a revised version of the manuscript that addresses the points raised during the review process.

We look forward to receiving your revised manuscript.

Kind regards,

Paul Hinckley Delano, Ph.D.

Academic Editor

PLOS ONE

Journal Requirements:

Additional Editor Comments :

Reviewer 1 has raised concerns regarding the ABR sample size (n=30) at 8 years old. It is crucial to address this limitation explicitly in the discussion section. Please add a caveat.

Reviewers' comments:

Reviewer's Responses to Questions

**Comments to the Author**

1. If the authors have adequately addressed your comments raised in a previous round of review and you feel that this manuscript is now acceptable for publication, you may indicate that here to bypass the “Comments to the Author” section, enter your conflict of interest statement in the “Confidential to Editor” section, and submit your "Accept" recommendation.

Reviewer #1: (No Response)

Reviewer #2: All comments have been addressed

2. Is the manuscript technically sound, and do the data support the conclusions?

Reviewer #1: Partly

Reviewer #2: Yes

3. Has the statistical analysis been performed appropriately and rigorously? 

Reviewer #1: Yes

Reviewer #2: Yes

4. Have the authors made all data underlying the findings in their manuscript fully available?

Reviewer #1: Yes

Reviewer #2: Yes

5. Is the manuscript presented in an intelligible fashion and written in standard English?

Reviewer #1: Yes

Reviewer #2: Yes

6. Review Comments to the Author

Reviewer #1: Dear Authors,

I would like to express my appreciation for your response to my previous comments on the revised version of your manuscript.

After careful consideration of the revisions made, particularly those addressing minor concerns, I find that there are still aspects that require attention, particularly with regard to the primary objectives related to the predictive value of neonatal click ABR in healthy children with respect to their perceptual auditory thresholds throughout life up to the age of 8 years.

As mentioned in my original report, the number of subjects at 8 years of age (n=30) appears to be low to fully address the objectives outlined in the article. Furthermore, the predictive value of ABR with behavioural thresholds is well established. Therefore, if the incidence of fluctuating hearing loss is low or excluded from the analysis, the predictive value will naturally remain high as long as the hearing level remains stable, regardless of age.

Under these circumstances, I regret to inform you that I am unable to recommend your manuscript for publication. However, as I mentioned in the initial report, I appreciate the reported time period of the extended cohort and the technical quality of the reported measurements. Once again, I recommend that the manuscript be reformulated towards a more descriptive approach. In such a case, it is the responsibility of the editorial team to determine whether the relevance and impact of the paper meet the requirements of PLOS ONE.

Best Regards

Reviewer #2: (No Response)

7. PLOS authors have the option to publish the peer review history of their article (what does this mean?). If published, this will include your full peer review and any attached files.

Reviewer #1: No

Reviewer #2: No

---

## [Author Response · Author response to Decision Letter 1]

27 Dec 2023

Dear Editor and Reviewers,

We extend our heartfelt appreciation to the reviewers and the editor for their invaluable insights and constructive comments. With the incorporation of the suggested modifications and additions, we hope to be reconsidered for publication. 

In the section below, we offer response to the comments received.

With kind regards,

Jolien Kleinhuis

Also on behalf of the other authors: 

Karin de Graaff, Paula van Dommelen, Irma van Straaten en Ruben Benard

Additional Editor Comments: Reviewer 1 has raised concerns regarding the ABR sample size (n=30) at 8 years old. It is crucial to address this limitation explicitly in the discussion section. Please add a caveat.

Answer: We have carefully considered the concerns about the ABR sample size at 8 years old and we understand the concerns of Reviewer 1. Therefore we made two major changes in the manuscript. 

First of all, we reformulated the manuscript towards a more descriptive approach, as Reviewer 1 suggested. We changed the objective in the abstract (line 20-21), in the introduction (line 81-88), in the method (line 102-104) and in the discussion (line 291-294).

Secondly, according to the suggestion of the Editor, we added a caveat to our discussion, aiming to explicitly address this limitation (line 349-352). These changes towards the descriptive approach resulted in a reformulation of the title (line 1) and the conclusion (line 360-363). 

Finally, we made minor adjustments in two sentences to enhance readability (line 157 and 313).

Overall, we believe these refinements contribute positively to the overall strength of this manuscript and we thank the Editor and Reviewer 1 for bringing this to our attention.

Comments to the Author

Reviewer #1: Dear Authors,

I would like to express my appreciation for your response to my previous comments on the revised version of your manuscript.

After careful consideration of the revisions made, particularly those addressing minor concerns, I find that there are still aspects that require attention, particularly with regard to the primary objectives related to the predictive value of neonatal click ABR in healthy children with respect to their perceptual auditory thresholds throughout life up to the age of 8 years.

As mentioned in my original report, the number of subjects at 8 years of age (n=30) appears to be low to fully address the objectives outlined in the article. Furthermore, the predictive value of ABR with behavioural thresholds is well established. Therefore, if the incidence of fluctuating hearing loss is low or excluded from the analysis, the predictive value will naturally remain high as long as the hearing level remains stable, regardless of age.

Under these circumstances, I regret to inform you that I am unable to recommend your manuscript for publication. However, as I mentioned in the initial report, I appreciate the reported time period of the extended cohort and the technical quality of the reported measurements. Once again, I recommend that the manuscript be reformulated towards a more descriptive approach. In such a case, it is the responsibility of the editorial team to determine whether the relevance and impact of the paper meet the requirements of PLOS ONE.

Best Regards

Answer: In the first place, we want to thank you for the effort you have made to bring your concerns about the article to our attention. We agree with you; our initial objective of the manuscript focused on the predictive value of the click ABR, but the limited sample size imposes constraints on the generalizability of our findings. 

Therefore, following your suggestions, we have made the document more descriptive. As mentioned in the response to the Editor as well, we changed the objective in the abstract (line 20-21), in the introduction (line 81-88), in the method (line 102-104) and in the discussion (line 291-294). Furthermore, we added a caveat to our discussion, aiming to explicitly address this limitation (line 349-352). These changes resulted in a reformulation of the title (line 1) and the conclusion (line 360-363).

Furthermore, thanks to your comments about the number of subjects, we made changes in description of the sample size (line 29-31 and 313) to clarify the number of children successfully finishing the 8-year follow-up. 

Your input has been of great value, and we hope these changes improve alignment with your expectations.

---

## [Editor Report · Decision Letter 2]

3 Jan 2024

An eight-year follow-up on auditory outcomes after neonatal hearing screening

PONE-D-23-17786R2

Dear Dr. Kleinhuis,

We’re pleased to inform you that your manuscript has been judged scientifically suitable for publication and will be formally accepted for publication once it meets all outstanding technical requirements.

Kind regards,

Paul Hinckley Delano, Ph.D.

Academic Editor

PLOS ONE
---

## [Editor Report · Acceptance letter]

19 Feb 2024

PONE-D-23-17786R2 

PLOS ONE

Dear Dr. Kleinhuis, 

I'm pleased to inform you that your manuscript has been deemed suitable for publication in PLOS ONE. Congratulations! Your manuscript is now being handed over to our production team.

Kind regards, 

on behalf of

Dr. Paul Hinckley Delano 

Academic Editor

PLOS ONE